# CHATREARRANGE: LEARNING TEXT-GUIDED 3D SCENE REARRANGEMENT

## ABSTRACT

In this paper, we propose ChatRearrange, a novel framework for the text-guided 3D scene rearrangement task. Instructing an algorithm with a text description to rearrange 3D furniture objects remains an unsolved problem in the 3D field. Unfortunately, developing algorithms to address this problem presents two critical challenges. First, we lack appropriate text-labeled scene data for the training procedure. Second, evaluating performance is challenging due to the absence of appropriate benchmarks. To address the first issue, we propose the ChatRearrange framework, which includes an LLM-based Inverse Distillation algorithm, enabling ChatRearrange to train without description-labeled scene data. Additionally, we incorporate a novel gradient-field-based student network to learn the text-3D knowledge from the LLM. For the second challenge, we benchmark the text-guided 3D scene rearrangement task by proposing a new dataset called TextRoom. We also include various metrics for the evaluation. The results show that our algorithm outperforms other baselines by a large margin. We are committed to releasing all the code and dataset if the paper is accepted.

## 1 INTRODUCTION

In our daily lives, when we move into a new house, rearranging our furniture objects into suitable positions is an extremely time-consuming task. Although we already have a blueprint in mind, it is still troublesome since we need to move the furniture objects to the desired positions. Imagine that if we have a robot that can correctly interpret our needs, and automatically rearrange the whole room, it will have a broader market in the field of home-use robotics. We demonstrate some examples of text description-guided automatic room rearrangement in Fig... In this figure, we send some instructions to the system, and it can give reasonable rearrangements that follow these descriptions.

This is a very difficult task for three main reasons. First, we need to build mechanisms for the algorithm to understand the meaning of the input text. Second, it must also comprehend the relationships among all the furniture objects and predict reasonable positions for these items based on the input text. Additionally, this task involves another critical, unsolved problem: how to map textual meaning to the real-world relationships of 3D objects in 3D space.

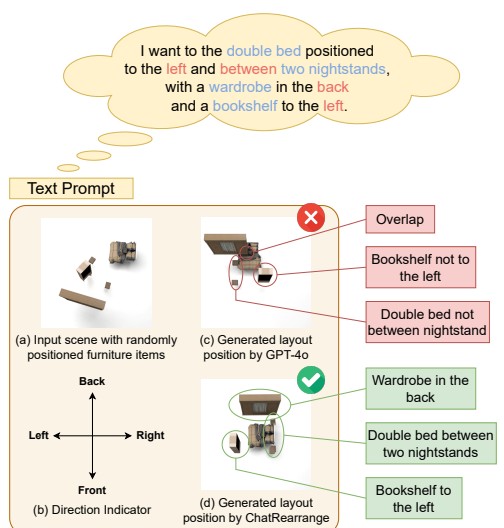

Figure 1: We ask GPT-4o and our proposed framework ChatRearrange to rearrange a messy room. The image above demonstrates that our proposed algorithm can produce a cleaned, rearranged room, while GPT-4o fails to achieve this goal.

Previous methods works have achieved some results on the topic of rearranging furniture objects or other 3D objects. Weihs et al. (2021) propose a new benchmark dataset called RoomR for the room rearrangement research. Their dataset includes 6,000 distinct rearrangement scenarios across 120 diverse scenes, including 72 different object types. Besides, they also establish baselines for benchmarking the task. Wang et al. (2023) introduces an optimization-based approach that considers the collaboration between the robots and humans. By utilizing Adaptive Simulated Annealing (ASA) and Covariance Matrix Adaptation Evolution Strategy, their method is able to optimize the human-robot co-activity for the rearrangement task. LEGO-Net Wei et al. (2023) is another research work for the room rearrangement task. Wei et al. (2023) use a diffusion-based approach Ho et al. (2020) to learn the rotation and translation. With this formulated learning objective, the model can predict the reasonable transformation for all the pieces of furniture and place them into the correct position.

However, although these methods can inspire us how to rearrange a mess room, we still face two big challenges when we implement an algorithm to achieve the goal of text-guided 3D room rearrangement. First, these methods do not consider the input of text descriptions. They basically only receive the shape information of the objects in a room, and predict their appropriate positions. We can only get a room with some random rearrangements, while we cannot control the behavior of the algorithms. In other words, we may not achieve our desired rearrangements through these discussed approaches. Secondly, we expect to use a large dataset with numerous text-labeled room data. However, the current datasets Weihs et al. (2021); Fu et al. (2021) in the room rearrangement area are difficult to fulfil this requirement, which brings another difficulty to build our text-guided algorithm.

We propose ChatRearrange framework to address the challenges mentioned above. Basically, our framework includes two parts: an LLM-based Inverse Distillation algorithm (LID) and a Text-guided 3D Rearranger with Gradient-Field Learning (TRGF). The function of LID is to address the problem of lacking text-labeled room data. Basically, we design two techniques in LID, including Rule-based Relation Understanding and 3D Scene Chain-of-Thought (3DSCoT). These techniques enable the LID to give accurate sentences with natural and human-like tones that can describe the relationships among all the furniture objects. TRGF plays the role of the student network, which receives knowledge from both the LID and the training dataset. With a successful distillation process, we can obtain a powerful Gradient-Field-based Model which can precisely sample suitable rearrangement plans based on the given texts. We use the proposed distillation to create 11166 training data in total.

In addition to the challenges discussed above, the evaluation of the text-guided 3D room rearrangement task can be difficult. First, we train our model with a distillation-based approach. In this case, we lack a suitable human-labeled dataset for the testing purpose. Therefore, we propose a dataset called TextRoom, which contains 319 text-labeled room scenes. The scenes are obtained from 3D-Front dataset Fu et al. (2021), and labeled with human efforts. Besides, the evaluation in the current research works Weihs et al. (2021); Wang et al. (2023); Wei et al. (2023); Tang et al. (2024) mainly focuses on testing the quality of the rearrangements, while it is very hard to apply these metrics to check how the rearrangements match with the text description. To address this issue, we propose a new metric called the Text-Scene-Matching Score (TSMS). This metric provides a more comprehensive evaluation by leveraging the power of the LLM, which can test whether the rearranged scene can match with the input text prompt.

**Contributions**

- We propose ChatRearrange framework to address the novel task text-guided 3D room rearrangement. Our framework includes a novel distillation policy - LID algorithm, which utilizes the 3D spatial knowledge from the LLM. This method enables our ChatRearrange can be trained by using 3D room data without text labels. We create 11166 training data in total using the proposed distillation policy.

- We also introduce a novel student network called TRGF. Our architecture leverages the power of the Gradient Field Song et al. (2020), language encoder and GNN to generate reasonable rearrangements for all the furniture objects.

- For fair comparison purposes, we benchmark the text-guided 3D room rearrangement task with a new testing dataset called TextRoom. This dataset contains 319 text-labeled room data points, which can be used for a comprehensive evaluation of our task. Besides, it

also has the potential to be applied in other tasks, such as 3D object detection, 3D scene generation, 3D relation interpretation, etc.

- We introduce a new metric called TSMS to evaluate how the rearranged scene matched the input text description in this task.

## 2 RELATED WORKS

### 2.1 3D FRONT DATASET

3D-FRONT Fu et al. (2021) is a dataset that contains 18,797 diversely furnished rooms across 6,813 houses, populated by over 13,151 high-quality textured 3D furniture objects. The dataset provides complete scene data from layout semantics to texture details, with professionally designed floorplans and expert-curated interior 3D-FRONT website designs ensuring style consistency. It enables data-driven design studies such as floorplan synthesis, interior scene synthesis, and scene compatibility prediction, while supporting 3D scene understanding research, including SLAM, reconstruction, and segmentation. The dataset serves as a foundational resource for advancing both generative modeling and scene understanding in computer vision applications. 3D-FRONT has become essential for numerous research efforts, including recent work like 3D-GRAND Yang et al. (2025) that uses it for training large-scale 3D language models. Other researchers leverage it for scene graph-to-3D synthesis research, compositional scene generation, and layout-shape generation methods Bahmani et al. (2023). It serves as a standard benchmark across indoor scene synthesis, layout generation, and 3D scene understanding literature, etc.

### 2.2 3D ROOM REARRANGEMENT

Recent studies have shown their success in 3D object rearrangement. Several works Gao et al. (2023); Gupta et al. (2023); Lou et al. (2023) focus on specific physical operations and path planning, considering object-to-object dependencies, travel efficiency, and trajectory optimization. However, these approaches require explicit user-defined targets. LEGO-Net, introduced by Wei et al. Wei et al. (2023), and DeBARA, proposed by Maillard et al. address rooms are capable of rearrangement with 3D reasoning without specific goal states, but they are not text-conditioned, limiting user control over the output pattern. Kapelyukh and Ren et al. proposed Dream2Real Kapelyukh et al. (2024), which integrates a vision language model (VLM) to select the rearrangement that best matches user instructions among multiple candidate positions in a dense and regular 3D grid. Yet, it primarily focuses on moving single objects to satisfy predefined relationships, lacking the capability for simultaneous multi-object rearrangement. Other works, like PanoGen Li & Bansal (2023), Chat2Layout Wang et al. (2024), Ctrl-RoomFang et al. (2024) generate text-conditioned 2D / 3D layouts, do not directly address the rearrangement of existing 3D objects within a scene.

### 2.3 LLM-BASED DISTILLATION

LLM-based distillation is a knowledge transfer technique that enables smaller "student" models to learn from larger "teacher" language models, preserving performance while dramatically reducing computational demands and deployment costs. The core principle operates on a teacher-student paradigm where a larger, more advanced model imparts its knowledge to a smaller, lightweight model by having the student observe and learn from the teacher's predictions, adjustments, and responses to various inputs. Some LLM distillation techniques often employ reverse Kullback-Leibler divergence rather than forward KLD, which is more suitable for generative language models and prevents the student from overestimating low-probability regions of the teacher distribution Gu et al. (2023). Advanced techniques like "distilling step-by-step" extract informative natural language rationales and intermediate reasoning steps from LLMs, enabling more data-efficient training Hsieh et al. (2023). The applications are extensive and transformative: enabling deployment of AI capabilities on resource-constrained devices and edge computing environments, domain specialization for fields like legal, financial, and medical applications, creating multilingual models and enabling on-device AI while circumventing privacy and computational complications, and facilitating self-improvement of open-source models by employing themselves as teachers Xu et al. (2024). This technique has become crucial for making advanced AI capabilities more accessible, efficient, and

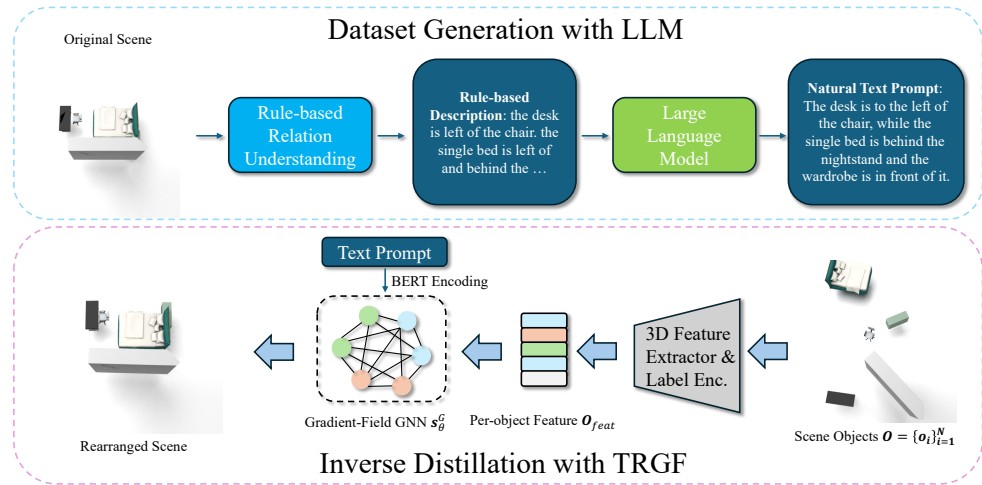

Figure 2: The pipeline of our proposed ChatRearrange framework. We train our model with an LLM-based Inverse Distillation (LID) manner. To be specific, we first use the proposed rule-based method to get the relations among all the objects. Then we use the LLM with our proposed 3DSCoT to transform the obtained relations into natural text descriptions.

practical for real-world deployment while maintaining the sophisticated reasoning abilities of larger models.

## 2.4 GRADIENT-FIELD LEARNING

Gradient-Field Learning Song et al. (2020), or score-based generative modeling, estimates high-dimensional probability distributions by learning the gradient of the log-density (the score function) instead of the density itself. Deep neural networks approximate $\nabla_{\mathbf{x}} \log p(\mathbf{x})$ using denoising score matching or similar objectives, capturing the geometry of the data manifold without explicit normalization. Sampling methods like Langevin dynamics or probability flow ODEs then generate new data from the learned distribution. This technique has proven effective across domains such as image and text-to-image generation, 3D modeling, audio synthesis, and molecular sampling, making it a cornerstone of modern generative modeling.

## 3 PROBLEM FORMULATION

Let $\mathbf{O} = \{\mathbf{o}_i\}_{i=1}^N$ denotes a set of furniture objects, where $\mathbf{o}_i = (\mathbf{p}_i, \mathbf{c}_i)$ represents a single object including the point cloud $\mathbf{p}_i \in \mathbb{R}^{2048 \times 3}$ and the corresponding class label $\mathbf{c}_i$. $\mathbf{U} = \{\mathbf{u}_i\}_{i=1}^N$ represents the corresponding set of furniture transformations, with each $\mathbf{u}_i$ aligned to the furniture object $\mathbf{o}_i$ of the same index. A single transformation $\mathbf{u}_i$ includes rotation and translation, which rotates and translates the 3D furniture into the correct posture and 3D position. Given a user's text description, the objective is to define a function $\hat{\mathbf{U}} = f(\text{Text}, \mathbf{O})$ that predicts reasonable furniture transformations $\hat{\mathbf{U}}$ satisfying the user's textual requirements.

## 4 METHOD

As stated above, our target is to obtain a function $\hat{\mathbf{U}} = f(\text{Text}, \mathbf{O})$. However, we have discussed in Sec. 1 that the dataset lacks text descriptions. In this case, we first need to obtain the corresponding text from the arranged rooms, and then we can train our text-guided 3D room rearrangement model. This motivates us to propose the LLM-based Inverse Distillation.

### 4.1 LLM-BASED INVERSE DISTILLATION (LID)

In the area of LLM Distillation, a typical distillation method is to use LLM for label annotating He et al. (2023); Xu et al. (2024). We have a dataset $X$, and our target is to use a smaller model to learn a mapping $f : X \rightarrow Y$. In this case, we can first use LLM to predict labels, and then we train the student model based on the generated labels.

Unfortunately, the label-annotating distillation method may not be suitable for our task. We need a special method to achieve our task. As stated before, our model needs to learn $\hat{\mathbf{U}} = f(\text{Text}, \mathbf{O})$. However, we require the LLM to generate the text descriptions for the rooms. In our design, the LLM predicts the text descriptions based on the given objects and the corresponding ground truth positions. Mathematically, the LLM plays the role of the function $\hat{\text{Text}} = \tilde{f}(\mathbf{U}, \mathbf{O})$, which is the inverse direction of the function $f$. For convenience, we name this special distillation policy as "inverse distillation".

In the following content, we introduce two important modules in our proposed LID, including Rule-based Relation Understanding and 3D Scene Chain-of-Thought (3DSCoT).

**Rule-based Relation Understanding**  Basically, we compare the direction and distance of different objects in the original scene to establish spatial relationships between them. This approach analyzes geometric positions and applies predefined spatial rules to determine relative orientations such as "left of," "right of," and "behind."

**3D Scene Chain-of-Thought (3DSCoT)**  We demonstrate our proposed 3DSCoT technique in Fig. 3. 3DSCoT (3D Scene Chain-of-Thought) aims to help language models convert repetitive, technical descriptions of furniture positions in rooms into natural, concise English descriptions while preserving spatial relationships. The method uses structured reasoning steps to guide the model through identifying furniture types, analyzing spatial relationships, and focusing on essential directional information. This allows it to convert technical spatial data into concise, natural-sounding descriptions that preserve important spatial relationships while being more accessible than the original format.

### 4.2 TEXT-GUIDED 3D REARRANGER WITH GRADIENT-FIELD LEARNING (TRGF)

The second important module of our proposed framework is TRGF, which is the student network of the inverse distillation process. We demonstrate the basic idea of TRGF in the lower part of Fig. 2. The function of TRGF is to learn a conditional probability distribution $p(\mathbf{U} \mid \text{Text}, \mathbf{O})$. To be specific, we learn the gradient field of log-conditional-density $\mathbf{s}_\theta = \nabla_{\mathbf{U}} \log(p_t(\mathbf{U} \mid \text{Text}, \mathbf{O}))$. We design several modules to achieve the estimation of the gradient field $\mathbf{s}_\theta$.

**Object Encoding**  The first step of TRGF is to encode the objects. For this purpose, we design a Label Encoder and apply a PointNet (the 3D feature extractor) Qi et al. (2017) to encode the object labels and point clouds, respectively. The Label Encoder is implemented with an MLP. We add the outputs from the networks to get the per-object features $\mathbf{O}_{feat}$.

**Text Encoding**  In addition to the object encoding, we also need to process the input text description. The encoded text $\text{Text}_{enc}$ is obtained through a text encoder, and we apply BERT Devlin (2018) to achieve the function of text encoding. With the per-object features $\mathbf{O}_{feat}$ and the encoded text $\text{Text}_{enc}$, we can then prepare our GNN nodes for the next step.

**Relation Learning with GNN**  In our daily lives, people often need to use relationships to describe their rearrangement requirements. For example, people may say, "Please put the table to the left of the double bed." In this case, it is quite important to learn the relationship among all the input objects. We propose to use a Gradient-Field GNN to achieve this goal. We first use the per-object features and encoded text to establish the GNN nodes. As demonstrated in Fig. 2, we concatenate $\text{Text}_{enc}$ to each object feature to get the corresponding node. Then, we build a Gradient Field GNN $\mathbf{s}_\theta^G = \nabla_{\mathbf{U}} \log(p_t(\mathbf{U} \mid \text{Text}_{enc}, \mathbf{O}_{feat}))$ to capture the relationship among all the nodes. Finally, we can sample from the trained GNN to obtain a suitable transformation $\hat{\mathbf{U}}$.

**Training**  The training of TRGF is a process of estimating the target conditional distribution $\nabla_{\mathbf{U}} \log(p_t(\mathbf{U} \mid \text{Text}, \mathbf{O}))$. We utilize the standard score-matching Song et al. (2020) training objective function to train our constructed model:

$$\mathcal{L} = \lambda(t) \|\mathbf{s}_\theta - \nabla_{\mathbf{U}(t)} \log p_{0t}(\mathbf{U}(t) \mid \mathbf{U}(0), \text{Text}, \mathbf{O})\|_2^2, \tag{1}$$

where $t$ is uniformly sampled from $[0, T]$; $\lambda$ is a function of weighting; $\mathbf{U}(t)$ and $\mathbf{U}(0)$ represent the perturbed and the origin transformation data respectively. In our training process, the Label Encoder and PointNet are co-trained with the Gradient Field GNN. We first use the text encoder to obtain $\text{Text}_{enc}$ and the Label encoder with PointNet to calculate $\mathbf{O}_{feat}$. Then we can get the score $\mathbf{s}_\theta$ through the Gradient Field GNN. Finally, we calculate the loss by using Eqn. 1. We repeat the training until the model is convergent.

**Inference**  We apply Predictor-Corrector Sampling algorithm Song et al. (2020) at the inference stage to obtain reasonable transformations. This sampler is an iterative process that repetitively uses the predictor and corrector to sample new data from the estimated Gradient Field. In our implementation, we apply Euler-Maruyama Song et al. (2020) as the predictor:

$$\mathbf{U}(t-\Delta t) = \mathbf{U}(t) + \sigma^{2t}\mathbf{s}_\theta \Delta t + \sigma^t \sqrt{\Delta t} \mathbf{z}_t, \tag{2}$$

where $\mathbf{z}_t \sim \mathcal{N}(\mathbf{0}, \mathbf{I})$. Basically, the predictor gives us the main direction of finding the target sample. To improve the sampling performance, we further use Langevin MCMC to make correction at each prediction step:

$$\mathbf{U}(t)_{i+1} = \mathbf{U}(t)_i + \epsilon \mathbf{s}_\theta + \sqrt{2\epsilon}\mathbf{z}_i, \tag{3}$$

where $\mathbf{z}_i \sim \mathcal{N}(\mathbf{0}, \mathbf{I})$, $\epsilon > 0$ indicates the step size of the correction. With the Predictor-Corrector Sampling algorithm, we can achieve high-quality results that fulfil the text description requirements.

> Instruction: You are given texts describing the positions of furniture in a room. The texts are repetitive and unnatural. Your task is to transform them into short and natural English descriptions that preserve the essential spatial relationships.
>
> Chain of Thought Reminder: Think through the text step by step, consider how a person would naturally describe the scene, then generate the most concise and human-sounding summary possible.
>
> 1. Read the text carefully.
> 2. Identify the main pieces of furniture. Please remember that some furniture types may contains multiple instances.
> 3. Use chain-of-thought reasoning to summarize the scene in a short, natural way.
> 4. Focus on essential directions (e.g. left, right, behind, in front) without sounding repetitive.
> 5. Provide the final concise descriptions.
>
> Finally, please remember you can only output one short and natural sentence to describe the room.

Figure 3: Our proposed 3D Scene Chain-of-Thought (3DSCoT). This instruction is applied to enhance the performance of the LLM to transform the previous rule-based descriptions into natural text prompts.

## 5 EXPERIMENTS

### 5.1 EVALUATION METRICS:

To evaluate and compare the performance of our model and baseline methods, we apply four quantitative metrics: Frechet Inception Distance (FID), Kernel Inception Distance (KID), Earth Mover's Distance (EMD), and Text-Scene-Matching Score (TSMS). These metrics are designed to assess the visual fidelity and semantic alignment of generated scene arrangements, either with ground truth layouts or corresponding textual descriptions.

**Frechet Inception Distance (FID)**  Originally designed for evaluating generative models, like GAN, FID measures the distance between two Gaussian distributions fitted to the features extracted from real and generated images by a pretrained Inception Network Heusel et al. (2017). It computes the Wasserstein-2 distance between these two distributions. In our setting, we measure the similarity between the rendered images of predicted layouts and their corresponding references. Lower FID values indicate higher fidelity and better consistency with the ground truth distribution.

**Kernal Inception Distance (KID)**  KID is a more statistically robust alternative to FID, based on the squared Maximum Mean Discrepancy (MMD) computed between real and generated image features in the feature space of the Inception Network Binkowski et al. (2018). Unlike FID, KID provides an unbiased estimator even with smaller sample sizes and is less sensitive to data scaling. In our context, KID measures the similarity between the rendered images of the predicted layouts and ground truth images. Lower KID scores reflect smaller distributional discrepancies.

**Earth Mover's Distance (EMD)**  EMD quantifies the minimal cost required to transform one spatial distribution into another Rubner et al. (2000). It is particularly suitable for tasks involving layout prediction and geometric shape, as it directly accounts for the spatial arrangement of objects. In our setup, we compute EMD between predicted and ground truth object bounding boxes by treating their locations as discrete spatial distributions. A lower EMD indicates that the predicted layout more closely matches the spatial configuration of the reference layout.

**Text-Scene-Matching Score (TSMS)**  The metrics discussed above quantify visual similarity (FID and KID) and geometric alignment (EMD) between the generated and ground truth layouts. However, they do not explicitly measure the semantic alignment between the input textual description and the generated layout. To address this gap, we propose the Text-Scene-Matching Score (TSMS), a new metric that directly evaluates how well the generated layout semantically matches the input text.

The computation of TSMS can be summarized as follows:

- **Scene-to-Text Conversion:** We first convert the generated 3D layout into rule-based textual representation. Several sentences are generated to describe the relationship between pairs of furniture sets.

- **Natural Language Generation:** The rule-based representation is then fed into an LLM, specifically GPT-4o, denoted as $\tilde{f}(\cdot)$, to generate a natural language description of the scene. Let $\mathbf{U}$ and $\mathbf{O}$ denote the spatial and object-level features extracted from the layout. The predicted textual description is given by:

$$\hat{\text{Text}}_{pred} = \tilde{f}(\mathbf{U}, \mathbf{O}) \tag{4}$$

  These two steps are similar to how we created the input text prompt.

- **Textual Similarity Measurement:** To compare the semantic alignment between the predicted $\hat{\text{Text}}_{pred}$ and the original input text prompt $\text{Text}_{gt}$, we encode both texts using Sentence-BERT Reimers (2019). Let $\phi(\cdot)$ denote the Sentence-BERT encoder, then the final TSMS is defined as the cosine similarity between the two encoded text vectors:

$$\text{TSMS} = cos(\phi(\hat{\text{Text}}_{pred}), \phi(\hat{\text{Text}}_{gt})) \tag{5}$$

A higher TSMS indicates a stronger semantic alignment between the generated layout and the original textual input.

These evaluation metrics measure the performance of candidate models from multiple perspectives, including visual fidelity (FID, KID), spatial accuracy (EMD), and semantic alignment with the input text (TSMS).

## 5.2 Results

### 5.2.1 Quantitative Results

We demonstrate the quantitative comparisons between our model and the baselines in Table 1. The result shows that our model consistently outperforms all baselines across all four evaluation metrics. The significantly lower FID and KID scores indicate that the visual appearance of our predicted layouts is more similar to the ground truth. The reduced EMD suggests that our model generates spatial transformations that more closely match the actual object positions. Additionally, the highest TSMS score reflects the superior semantic alignment between our generated layout and the input text prompts.

### 5.2.2 Qualitative Results

We present the qualitative comparisons of our model and other baselines in Figure 4. It is important to note that not all furniture items are explicitly mentioned in the text prompts, and even when they are, their precise positions may not be fully specified. Therefore, the models are expected not only to correctly interpret the described spatial relationships between the mentioned furniture pairs but also to infer suitable positions for the unspecific items based on the latent features.

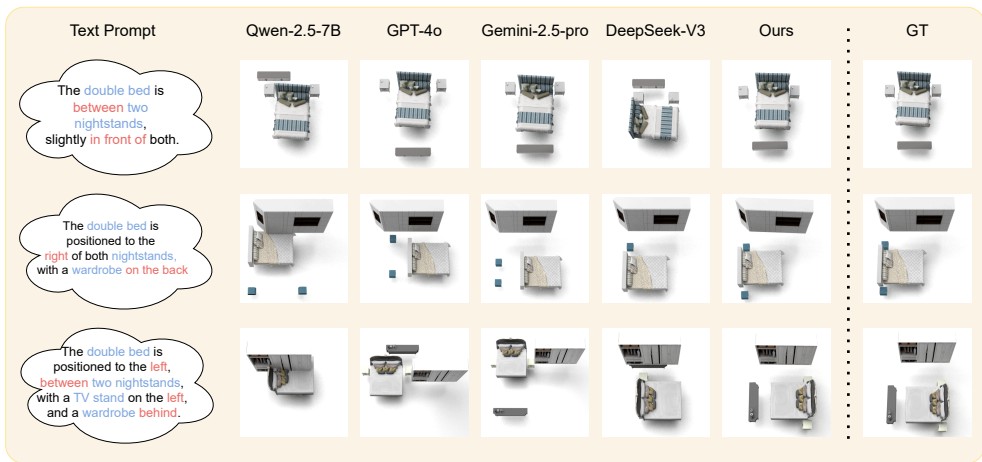

Figure 4: Qualitative comparison between LLM baselines and our model. Each row represents a single task. The first column shows the input text prompt, where furniture-related terms are highlighted in blue and direction-related terms are colored in red. The ground truth is shown in the last column. Our model captures both the latent and explicit spatial relationships described in the text, generating layouts that most closely align with the input descriptions. In contrast, the baseline models often fail to preserve key spatial relationships and semantic coherence. For Direction Indicator, please refer to Fig. 1. We present more qualitative results in Fig. 5 in Appendix.

For example, in the first scene, the TV stand is not mentioned in the text prompt. Both DeepSeek and Qwen failed to recognize that, under typical spatial conventions, the TV stand should be placed in front of the bed (at the foot of the bed). Furthermore, DeepSeek does not accurately interpret the phrase "double bed between two nightstands"—it aligns the bedside of the bed with the nightstands, rather than placing the head of the bed between them, as is semantically intended.

In the second scene, our model is the only one that correctly positions the two nightstands on opposite sides of the bed, properly aligned with the headboard. While GPT and Gemini do place the nightstands

Table 1: Quantitative comparison between baseline models and our proposed method. Our model achieves the best performance across all four metrics.

[†] KID scores are scaled by $10^3$
[1] Qwen-2.5-7B  [2] DeepSeek-V3
[3] Gemini-2.5-pro

| | FID ↓ | KID ↓ [†] | EMD ↓ | TSMS ↑ |
|---|---|---|---|---|
| Qwen[1] | 82.7 | 16.4 | 2.24 | 85.9 |
| GPT-4o | 84.1 | 15.5 | 2.12 | 82.1 |
| DS-V3[2] | 82.4 | 14.7 | 2.06 | 82.8 |
| Gemini[3] | 73.6 | 6.35 | 2.02 | 85.2 |
| Ours | **54.7** | **-0.685** | **1.68** | **88.9** |

symmetrically, they fail to correctly position the bed between them, resulting in misaligned configurations.

The third scene contains the most detailed description and the largest number of furniture items, all of which are explicitly mentioned in the prompt. Qwen struggles to handle the complex set of relationships, resulting in significant overlaps and incorrect placements. In contrast, our model is the only one that satisfies all the described spatial relationships, producing a layout that closely matches the ground truth.

These results show that our model not only capable of accurately capturing all explicit spatial conditions but also understanding the latent indications based on the furniture labels.

### 5.2.3 ABLATION STUDY

The ablation study on sampler selection and the sampling step is presented in Table 2. Considering all four evaluation metrics, the Predictor-Corrector (PC) sampler consistently outperforms the Euler-

Table 2: Ablation study on samplers (EM and PC) and sampling steps.

$^{\dagger}$ KID scores are scaled by $10^3$

| Sampler | EM | | | | PC | | | |
|---|---|---|---|---|---|---|---|---|
| Step | FID ↓ | KID $^{\dagger}$ ↓ | EMD ↓ | TSMS ↑ | FID ↓ | KID$^{\dagger}$ ↓ | EMD ↓ | TSMS ↑ |
| 100 | 100.0 | 33.2 | 1.91 | 84.1 | 99.8 | 34.9 | 1.95 | 85.2 |
| 200 | 73.1 | 12.4 | 1.72 | 87.6 | 72.3 | 12.2 | 1.67 | 87.8 |
| 300 | 61.2 | 3.32 | 1.70 | 87.9 | 59.9 | 2.18 | 1.69 | 88.3 |
| 400 | 57.5 | 0.692 | **1.65** | 88.1 | 56.9 | -0.124 | 1.66 | 88.2 |
| 500 | 56.4 | -0.108 | 1.67 | 88.0 | 55.3 | -0.692 | 1.68 | 88.6 |
| 600 | 56.5 | -0.378 | 1.68 | 88.3 | **54.7** | **-0.685** | 1.68 | **88.9** |

Maruyama (EM) sampler Song et al. (2020). This is expected, as the PC sampler introduces an additional corrector phase based on Langevin dynamics that refines each prediction step, reducing the error produced by the SDE solver Song et al. (2020).

Regarding the number of sampling steps, performance becomes stable after 400 steps, with the best results observed at 600 steps. This suggests that a higher number of sampling iterations contributes to more precise rearrangement, and the improvement plateaus beyond certain steps.

## 5.3 CAN LARGE MODELS DIRECTLY REARRANGE THE ROOMS?

As shown in both quantitative results, Table 1, and qualitative evaluations, Figure4, larger-scale models tend to perform better, and models augmented with reasoning capabilities (e.g. Gemini) demonstrate relative improvements.

While LLMs provide a natural interface for users to describe desired layout using textual input, they are facing three significant limitations:

- **Failure to capture implicit relationships:** LLMs often fail to consider latent spatial conventions embedded in furniture semantics. For example, models may place a TV stand in arbitrary locations, ignoring common-sense pairings such as placing it opposite the bed
- **Difficulty following all explicit spatial constraints:** When prompts contain multiple spatial relationships, especially involving many objects, LLMs struggle to satisfy all conditions simultaneously, i.e. Scene 3. This leads to degraded layout quality and misalignment between the text prompt and the generated arrangements.
- **Neglect of furniture volume and physical constraints:** LLMs often ignore the actual size, resulting in overlaps or other impractical configurations. For example, Qwen places the wardrobe over the double bed in some scenes.

In summary, while LLMs offer a promising interface to interact with human preferences via natural language, they lack the precise spatial reasoning needed for reliable room rearrangement. Our model addresses these issues with a Gradient-Field GNN, enabling realistic and closely-matched scene rearrangement.

## 6 CONCLUSION

We propose ChatRearrange framework for text-guided 3D scene rearrangement. It leverages LID to distill spatial knowledge from LLMs and trains a student network, TRGF, without human-annotated text. We introduce the TextRoom benchmark with 319 text-labeled 3D scenes. Our model outperforms LLM-based baselines across standard metrics and the proposed TSMS, showing strong potential for 3D scene rearrangement from textual input.

**Future Work**   Currently, we conduct our experiments in a virtual environment. In the future, we plan to implement our proposed pipeline in real-world scenarios, such as robotic applications. It will be interesting to see whether our algorithm can help robots understand the furniture in a room.

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

## A Appendix

### A.1 Code and Data Sample

For reproducibility, the implementation and data sample are provided in the anonymous repository. Please refer to `Code/README.md` for detailed setup, data usage, and model training and testing instructions.

### A.2 Qualitative Results

More qualitative results are presented in Fig. 5 (in the next page). Each row represents a single task with the text input in the first column. The furniture-related terms are highlighted in blue, and orientation-related descriptions are colored in red. Our model's predicted layout aligns the best with the text input and the ground truth. However, the baselines often fail to preserve all the described spatial relationships and latent features contained in the furniture labels. These results suggest our model has promising performance in the text-guided 3D scene rearrangement task.

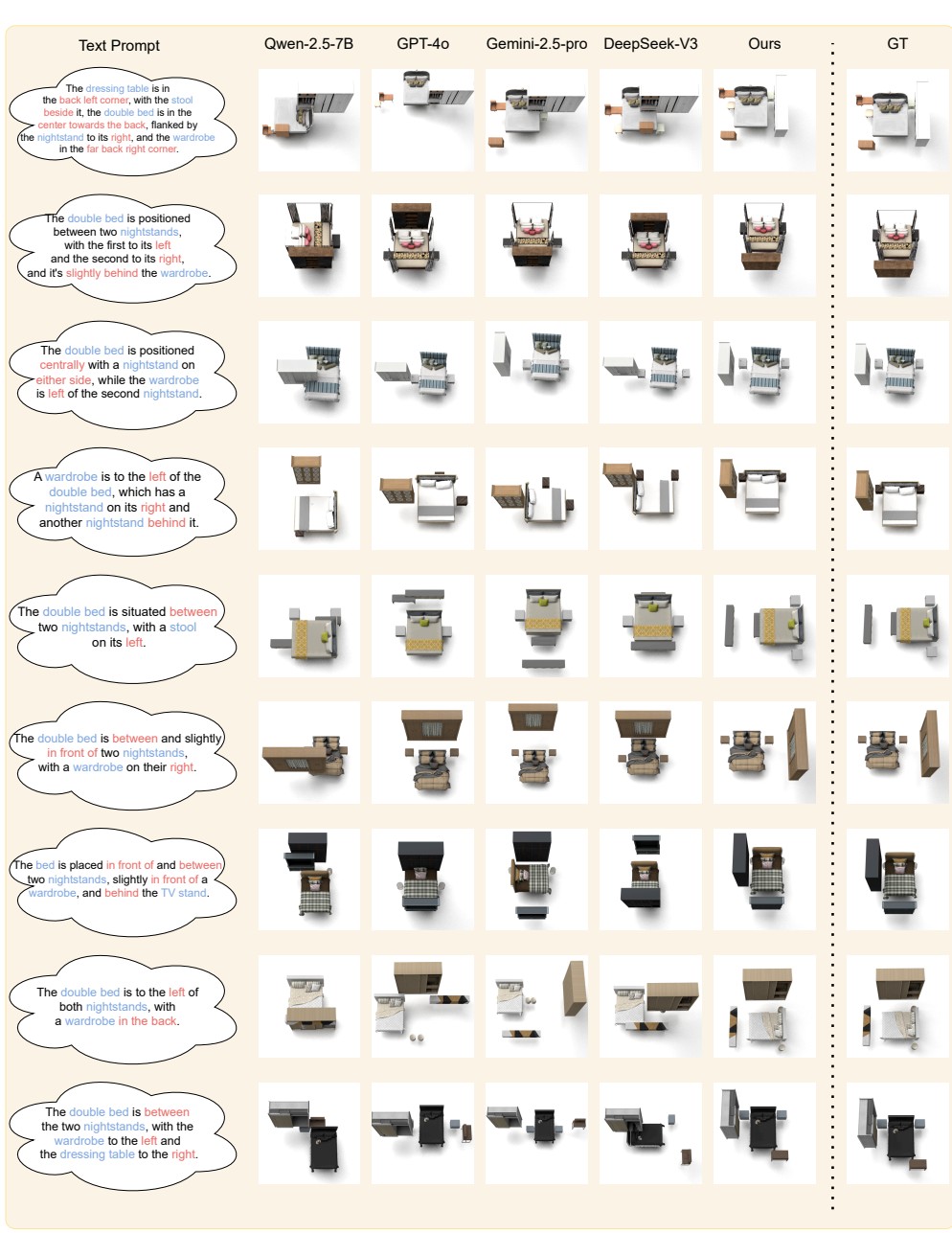

Figure 5: Qualitative comparison between LLM baselines and our model. Each row represents a single task. The first column shows the input text prompt, with furniture-related terms highlighted in blue and direction-related terms colored in red. The ground truth is shown in the last column. Our model captures both the latent and explicit spatial relationships described in the text, generating layouts that most closely align with the input descriptions. In contrast, the baseline models often fail to preserve all key spatial relationships and semantic coherence.

