# OpenReview forum: "ChatRearrange: Learning Text-guided 3D Scene Rearrangement"
_ICLR.cc/2026/Conference — ICLR 2026 Conference Withdrawn Submission_

### Official Review · Reviewer_TPLT · 2025-10-26

**Soundness:** 4
**Presentation:** 3
**Contribution:** 4
**Rating:** 4
**Confidence:** 4

**Summary:**

This paper introduces ChatRearrange, a novel framework designed to rearrange 3D furniture objects within a scene according to a natural language text prompt. This paper proposes an LLM-based Inverse Distillation (LID) pipeline. Instead of labeling messy scenes with text, this method starts with well-arranged ground-truth scenes and uses an LLM to generate corresponding natural text descriptions. This process is guided by a 3D Scene Chain-of-Thought (3DSCoT) technique to ensure high-quality text.

**Strengths:**

1. The paper tackles a well-defined, practical, and challenging task. Text-guided rearrangement has clear applications in robotics and design , and the paper does a good job of differentiating it from related tasks like scene generation or unconditional rearrangement.

2. The core challenge for this task is the lack of labeled data. The proposed "inverse distillation" is a good solution. Generating text from existing high-quality scenes is more scalable than trying to manually create (messy scene, text prompt, clean scene) triplets. The use of 3DSCoT to refine rule-based descriptions into natural language is a solid contribution.

3. The choice of a score-based generative model (Gradient-Field Learning) is well-suited for this conditional generation task. Using a GNN to model the inter-object spatial relationships is a very natural fit, as rearrangement is fundamentally about these relationships (e.g., "bed between nightstands").

**Weaknesses:**

1. The primary weakness is the choice of baselines. The paper compares ChatRearrange against large language models (GPT-4o, Gemini, etc.). However, these are general-purpose text models, not specialized 3D geometry models. The paper itself argues (and shows) they are bad at this task. A much stronger comparison would have been against other 3D rearrangement methods (like LEGO-Net , mentioned in the related work ) adapted for text-conditioning. The paper dismisses these as not being text-conditioned  but doesn't attempt to create a stronger baseline by, for example, conditioning their latent space on the BERT text embedding. This makes the comparison not very solid.

2. The examples provided are relatively simple, involving a small number of objects (e.g., a bed, two nightstands, a wardrobe). It is unclear how the method would scale to more complex scenes with many objects or more complex, compositional text prompts (e.g., "Put the desk by the window, place the chair in front of it, and arrange the three bookshelves along the opposite wall").

**Questions:**

1. The paper is vague on how the LLM baselines (GPT-4o, Gemini) were made to perform this task. How was the 3D scene represented as input, and how were 3D coordinates/transformations decoded from their text output? This detail is critical for understanding and reproducing the comparisons in Table 1.

2. Could the authors provide an ablation study that trains the TRGF model only on the 319 human-labeled TextRoom samples? This would directly quantify the value and necessity of the 11,166-sample synthetic dataset generated by LID.

---

### Official Review · Reviewer_ECZi · 2025-10-29

**Soundness:** 2
**Presentation:** 2
**Contribution:** 2
**Rating:** 2
**Confidence:** 4

**Summary:**

The paper proposes ChatRearrange, a pipeline for text-guided 3D scene rearrangement. It tackles the lack of text-labeled training data via an LLM-based Inverse Distillation (LID) procedure that converts ground-truth object layouts into natural language using rule-based relation extraction plus a “3D Scene Chain-of-Thought” prompt. A Text-guided Rearranger with Gradient-Field Learning (TRGF) then learns a conditional score over object transformations given objects and text. The authors also introduce TextRoom and a Text-Scene-Matching Score (TSMS) that converts layouts back to text with GPT-4o and measures cosine similarity with Sentence-BERT. They report 11,166 synthetic training pairs and strong gains over LLM baselines on FID/KID/EMD/TSMS.

**Strengths:**

1. LID is a reasonable twist on distillation (inverse mapping Text≈f̃(U,O)) and is clearly motivated.
2. Conditional gradient-field over transformations with a GNN is technically sound and fits multi-object constraints.

**Weaknesses:**

1. Baselines are weakly matched: Main comparisons are to general-purpose LLMs prompted to place furniture (Qwen, GPT-4o, Gemini, DeepSeek). Other existing text-to-3D-layout baselines like InstructScene, EchoScene, LayoutGPT, and LayoutVLM are not included in the paper.
2. Lack of related works: Some recent related works that focus on scene synthesis and rearrangement are not discussed, especially for those works that also used GNN-based methods for scene layout generation, like InstructScene and EchoScene.
3. Limited test size/scope: TextRoom has only 319 scenes from 3D-FRONT; it’s unclear how diverse the prompts are, whether rooms/categories overlap with training, and whether generalization holds for other room types or open-vocabulary objects.
4. Missing implementation detail for reproducibility. The paper omits many specifics: GNN architecture/topology, noise schedule, σ_t, λ(t), PointNet/BERT variants and dims, relation rule thresholds, and collision/overlap handling.
5. Circularity of the TSMS Metric: TSMS converts layouts back to text with GPT-4o and then compares to the input via Sentence-BERT; this risks style leakage (training text is also produced with an LLM after a rule stage) and couples evaluation quality to a black-box vendor model. Human judgments or relation-level precision/recall would strengthen claims.

**Questions:**

1. Can the authors provide more detail on the "Rule-based Relation Understanding" module? What specific spatial predicates (e.g., "left of," "behind") are supported? How does it handle relationships between more than two objects, or more abstract/non-axial concepts?
2. Can you adapt text-to-layout methods to your setting and compare?
3. Why BERT over vision-language encoders (e.g., CLIP-style) for text? Did you try instruction-tuned text encoders?

---

### Official Review · Reviewer_q1VC · 2025-11-01

**Soundness:** 2
**Presentation:** 2
**Contribution:** 2
**Rating:** 2
**Confidence:** 4

**Summary:**

This paper proposes ChatRearrange, a framework for the text-guided 3D scene rearrangement task. It aims to address the lack of text-labeled training data and the absence of evaluation benchmarks by introducing an LLM-based Inverse Distillation (LID) algorithm to generate training data and a Gradient-Field-based student network (TRGF) to learn spatial relationships.

**Strengths:**

This paper presents a new benchmark dataset, TextRoom, and a new evaluation metric, TSMS, for semantic alignment. The idea of ChatRearrange demonstrates improved control over text-conditioned scene rearrangement.

**Weaknesses:**

1. The first motivation presented in the paper, i.e., “the lack of text-labeled scene data” is not convincing. Utilizing Vision-Language Models (VLMs) to generate object-level textual annotations is a well-explored direction in the literature, and several works such as InstrcutScene have successfully demonstrated this. The claim that this is a major obstacle does not hold, and the motivation for the proposed method is not sufficiently convincing.

2. The proposed method employs a score-matching objective, which is closely related to diffusion models. However, the authors do not compare their approach with any existing diffusion-based scene generation or rearrangement methods. Given the similarity in formulation, this omission significantly weakens the contextualization of the proposed method within the broader generative modeling landscape.

3. The role of 3DSCoT is unclear. Intuitively, transforming hard-coded spatial relationships into natural language descriptions does not require a Chain-of-Thought (CoT) mechanism. The authors introduce 3D Scene Chain-of-Thought (3DSCoT) without providing sufficient details or justification. The necessity and contribution of this component are unclear, and its implementation is also not adequately explained.

4. The proposed model does not clearly embody the concept of a "student" network in the traditional knowledge distillation sense. Instead, it appears to learn spatial and relational priors directly from the dataset. However, the dataset itself is not sufficiently described, e.g., its structure, statistics, and how it supports the learning of these priors are largely omitted, making it difficult to assess the validity of this approach.

5. Although the authors claim to introduce a new dataset (TextRoom) for evaluation, the paper provides minimal information about its construction, annotation process, or diversity. At the same time, there is also no explanation of how this dataset is used as a benchmark to validate model performance.

**Questions:**

In the problem formulation section Line218-228, the authors define the input as object set O and the goal as predicting transformations U. However, the variable Y, which presumably represents the text labels or descriptions, is introduced without a clear definition or role in the formulation. This omission creates confusion about the exact learning objective and weakens the clarity of the proposed framework.

---

### Official Review · Reviewer_zrXR · 2025-11-02

**Soundness:** 2
**Presentation:** 2
**Contribution:** 2
**Rating:** 2
**Confidence:** 3

**Summary:**

The paper proposes ChatRearrange, a novel framework for text-guided 3D scene rearrangement using LLM-based Inverse Distillation and gradient-field learning. It trains without human-annotated text descriptions and introduces the TextRoom benchmark with 319 labeled scenes. Experimental results demonstrate superior performance over LLM baselines across all metrics.

**Strengths:**

1. The paper is well written and easy to follow.

2. The proposed dataset could be helpful to some domains (e.g. 3D object detection, 3D generation/understanding).

**Weaknesses:**

1. The novelty is limited. The pipeline combines established techniques.

2. The experiment is limited. It only compares LLM-based methods. The paper lacks comparisons with existing learning-based approaches for scene rearrangement or layout generation, such as LEGO-Net, diffusion-based scene synthesis methods, or other neural architecture designs. Without these comparisons, it is difficult to assess whether the proposed gradient-field GNN architecture provides advantages over alternative learning-based designs.

3. The proposed metric TSMS relies on GPT-4o to convert layouts back to text, then uses Sentence-BERT for similarity. This creates circular dependency: using an LLM to evaluate what was generated using LLM distillation.

**Questions:**

Is there any error during LLM labeling? If yes, how do you handle it?

---

### Note · Authors · 2025-11-12

I have read and agree with the venue's withdrawal policy on behalf of myself and my co-authors.